# Common Practice in the Treatment of Superficial Vein Thrombosis Involving the Sapheno-Femoral Junction: Results from a National Survey of the Italian Society of Angiology and Vascular Medicine (SIAPAV)

**DOI:** 10.3390/medicina59061068

**Published:** 2023-06-01

**Authors:** Giuseppe Camporese, Pierpaolo Di Micco, Marcello Di Nisio, Walter Ageno, Romeo Costanzo Martini, Paolo Prandoni

**Affiliations:** 1General Medicine Unit & Thrombotic, and Haemorrhagic Disorders Unit, Department of Internal Medicine, University Hospital of Padua, 35128 Padua, Italy; 2AFO Medicine, PO Santa Maria delle Grazie Hospital, ASL Napoli 2 Nord, 80078 Napoli, Italy; pierpaolo.dimicco@aslnapoli2nord.it; 3Department of Medicine and Ageing Sciences, “G D’Annunzio” University, 66100 Chieti, Italy; marcello.dinisio@unich.it; 4Department of Medicine and Surgery, University of Insubria, 21100 Varese, Italy; walter.ageno@uninsubria.it; 5Unit of Vascular and Endovascular Surgery, San Martino Hospital, 32100 Belluno, Italy; martiniromeo@gmail.com; 6Arianna Foundation on Anticoagulation, 40100 Bologna, Italy; prandonip@gmail.com

**Keywords:** superficial vein thrombosis, treatment, anticoagulants, fondaparinux

## Abstract

*Background and Objectives*: Prophylactic doses of low-molecular-weight heparins or fondaparinux showed their efficacy and safety for treatment of all superficial vein thrombosis (SVT) of the lower limbs, yet not for those extended to the last 3 cm of the great saphenous vein, close to the sapheno-femoral junction, or considered as a deep-vein thrombosis. Some experts suggest that these patients should be managed with full anticoagulant doses but evidence to support this recommendation is lacking, suggesting the need for a properly designed trial. *Materials and Methods*: Before starting a new trial, the Italian Society of Angiology and Vascular Medicine (SIAPAV) decided to verify the common therapeutic approaches for patients with an SVT in Italian vascular centers based on a hypothetical significant variation in each daily clinical practice. A standardized questionnaire of 10 questions was administered to all SIAPAV affiliates by means of the official Society website. *Results*: From 1 December 2022 to 20 January 2023 a total of 191 members (31.8%) answered the questionnaire, showing a detailed and a substantial heterogeneity in the therapeutic approach to SVT patients among experienced vascular physicians and angiologists. Detailed results are reported in the relative section. *Conclusions*: The therapeutic approach of SVT extended to the iuxta-femoral segment of the great saphenous vein is still a matter of debate, and data to support therapeutic strategies are lacking. The wide heterogeneity in the management of SVT patients, including those with more extended thrombosis, confirmed that a randomized controlled clinical trial investigating the efficacy and the safety of a tailored therapeutic regimen in this particular subgroup of patients is strongly warranted.

## 1. Introduction

Superficial vein thrombosis (SVT) is a common entity encountered in daily clinical practice, with an incidence ranging from 0.3% to 1.5% per 1000 person years, similar to or even higher than that of deep-vein thrombosis (DVT) [1,2]. For a long time, SVT has been considered as a rather benign disease, predominantly associated with limited rather than systemic implications. However, several recent studies have shown a strong association with concomitant extension to the deep veins of the lower limbs and with pulmonary embolism (PE), mainly when SVT involves the great saphenous vein above the knee. Indeed, approximately 20% to 25% of patients with an acute SVT have a concomitant DVT and approximately 7% to 13% have a concomitant PE at the exact time or within the first 3 months from the objective SVT diagnosis [1,3,4].

A number of randomized controlled trials have shown the efficacy and safety of some parenteral anticoagulants, such as low-molecular weight heparins or fondaparinux administered at intermediate or prophylactic doses, and of a direct oral anticoagulant, such as rivaroxaban, administered at prophylactic doses, for about 6 weeks for the treatment of SVT [5,6,7].

Based on the results of these trials, this approach is recommended for most patients with SVT, with the exception of those with an extension of the thrombus within the last 3 cm from the sapheno-femoral junction. These patients were excluded from trials on SVT treatment, because of a perceived higher risk of venous thromboembolic complications, but were also excluded from trials on DVT treatment, because this site of thrombosis is not strictly considered as involving the deep venous system. Some experts suggest that these patients should be managed with full therapeutic anticoagulation, similar to those with a DVT, but evidence to support this recommendation is lacking [8]. Unfortunately, recent international guidelines on the treatment of venous thromboembolism do not include any recommendation on the management of these patients [9,10].

A recent paper reported a subgroup analysis of 374 patients with SVT involving the sapheno-femoral junction enrolled in the RIETE registry. These patients were managed with either therapeutic (n = 227, 60.7%) or prophylactic (n = 147, 39.3%) doses of anticoagulants at the discretion of attending physicians. The study showed a non-statistically significant reduction in the rates of venous thromboembolic complications, defined as SVT extension or recurrence, DVT or PE, in patients receiving therapeutic doses (odds ratio [OR] 0.48; 95% CI, 0.11 to 2.17; *p* = 0.33), and a non-statistically significant increase in major or clinically relevant non-major bleeding (OR, 2.0; 95% CI, 0.20 to 19.41; *p* = 0.56) [11].

These findings suggested the need for a properly designed trial aimed at better defining the optimal management strategy for patients with an SVT involving the last 3 cm of the great saphenous vein close to the sapheno-femoral junction. Meanwhile, the Italian Society of Angiology and Vascular Medicine (SIAPAV) decided to explore which were the most common therapeutic approaches for these patients at Italian member centers, based on the hypothesis that the lack of guidelines could have generated a significant variation in daily clinical practice.

## 2. Materials and Methods

A standardized questionnaire consisting of 10 predefined questions was prepared by a panel of 4 experts (P.P., G.C., M.D.N, P.D.M.), all members of the Working Group on Venous Thromboembolism of the SIAPAV Research Center.

All members of SIAPAV affiliates were invited to participate in the survey, which was made available on the official website of the Society. Each question was multiple choice and participants had to choose only one option corresponding to their actual clinical practice. All individual responses were collected anonymously and entered into a specific database (Survey Monkey Inc. San Mateo, CA, USA) which was qualified for online survey development. The list of the 10 questions sent to the participants is reported in Table 1. The results were analyzed using descriptive statistics, and data were reported as numbers and related percentages as well as plotted in diagrams.

## 3. Results

From 1 December 2022 to 20 January 2023, a total of 191 members (31.8% of the total number of SIAPAV members) answered the questionnaire. The affiliation and the specialty of the respondents are reported in Table 2.

Altogether, 8% of them reported managing less than 10 patients with a SVT per year, 58% 10 to 25 patients, and 34% more than 50 patients per year. Seventy-two percent of clinicians declared that SVT involving the last 3 cm of the great saphenous vein represents only 5–10% of the total cases of SVT diagnosed in their daily clinical practice; while 22% declared that these forms of SVT represent 10% to 20% of all diagnosed SVT; and only 6% declared to detect more than 20% of these SVT cases. The majority of clinicians (65%) prescribe fondaparinux 2.5 mg once-daily for the treatment of patients with SVT; 75% treat SVT for a total of 45 days. Of interest, some of the remaining respondents decided about duration and dose of anticoagulation based on the results of periodic ultrasonographic reassessment. In particular, while 106 clinicians (56%) maintained the same dosing regimen throughout the entire treatment course, 32 (17%) decided on whether to continue or withdraw anticoagulation based on the results of serial ultrasound scans; and 44 (23%) used ultrasonographic reassessment to decide on dose adjustments. It is worth mentioning that only one-third of clinicians adopted the same therapeutic regimen for all patients, whereas 67% of them tailored the regimen according to patients’ comorbidities. 

Most clinicians (89%) considered an SVT extended to less than 3 cm from the sapheno-femoral junction as a DVT (Q8, Figure 1), and 76.6% prescribed a full therapeutic dose of anticoagulants as for a proximal DVT, while 14.6% of them adjusted the dose according to the results of serial ultrasound scans (Q9, Figure 2). One hundred ten clinicians (57%) prescribed at least a 3-month course of anticoagulation, while fifty-five (29%) decided treatment the duration of treatment on the basis of a serial ultrasound control (Q10, Figure 3).

## 4. Discussion

The results of our survey show a substantial heterogeneity in the therapeutic approach to patients with SVT among experienced vascular physicians and angiologist members of SIAPAV.

Three important findings emerged from the survey: (1) the vast majority of respondents use parenteral anticoagulants (mainly prophylactic doses of fondaparinux) rather than direct oral anticoagulants for the treatment of SVT; (2) none of the clinicians prescribed vitamin K antagonists; and (3) two-thirds of clinicians appropriately prescribed anticoagulation for a period of 45 days, as recommended by the latest ACCP guidelines on VTE treatment [9]. Nevertheless, as many as 20–25% of clinicians adopted a different therapeutic strategy and treated SVT with different drugs, regimens, and treatment durations guided by imaging, location and extension of the thrombus, and patients’ comorbidities, which may eventually increase costs and utilization of health care resources. Similarly, substantial heterogeneity emerged with regard to the management of iuxta-femoral SVT (i.e., SVT involving the last 3 cm of the great saphenous vein and the sapheno-femoral junction), in particular with regard to the anticoagulant drug dose and to the optimal duration of therapy. However, even if 89% of clinicians considered iuxta-femoral SVT as a DVT, only 57% of them actually treated it with full-dose anticoagulants for at least 3 months, despite the lack of evidence to support such an approach and the absence of specific recommendations from the most important international guidelines [9,10]. We are aware of only one expert opinion paper in which the author suggested therapeutic doses of anticoagulant drugs as for DVT in patients with a proximal SVT extension < 3 cm to the sapheno-femoral junction. However, the same author acknowledges that the proposed treatment algorithm has never been formally validated [8].

We need to acknowledge that surgical procedures have been deliberately excluded from the questionnaire as potential therapeutic strategies. Surgery plays an uncertain role in the management of SVT, and the two most recent versions of the ACCP guidelines in 2016 and 2021 did not provide any suggestion or recommendation for surgery in SVT patients [9,12].

## 5. Conclusions

The therapeutic approach for SVT extended to the iuxta-femoral segment of the great saphenous vein involving the common femoral vein is still matter of debate, and data to support therapeutic strategies are lacking. The results of this survey showed heterogeneity in the overall management of SVT patients and, as expected, this also applies to patients with a more extended form of thrombosis, although the majority of respondents considered SVT extended to the iuxta-femoral segment as being more similar to DVT than to other forms of SVT.

A randomized controlled clinical trial investigating the efficacy and the safety of tailored therapeutic regimen in this particular subgroup of patients is strongly warranted.

## Figures and Tables

**Figure 1 medicina-59-01068-f001:**
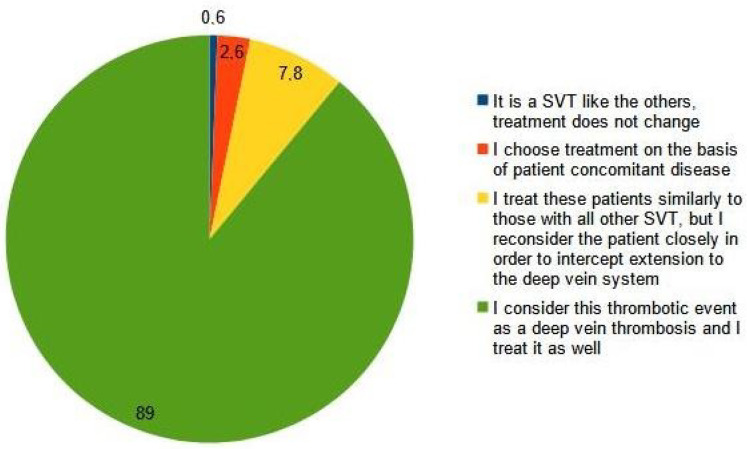
Q8: Which is your therapeutic strategy for the treatment of a superficial vein thrombosis involving the last 3 cm of the great saphenous vein or of the small saphenous vein and extended close to the sapheno-femoral or sapheno-popliteal junctions, respectively?

**Figure 2 medicina-59-01068-f002:**
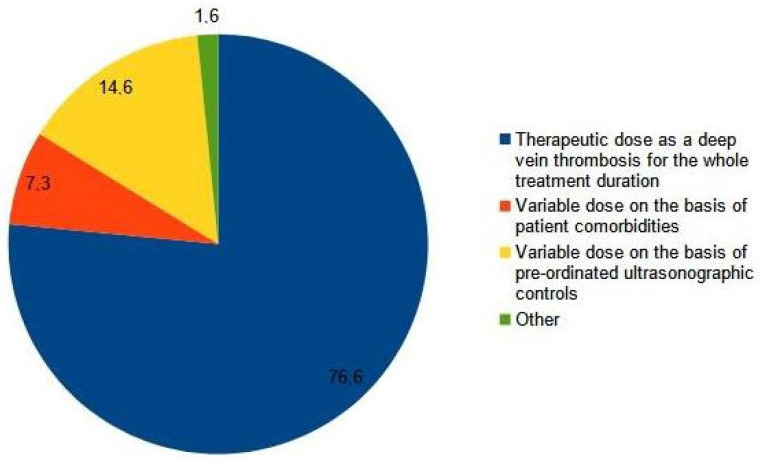
Q9: You prescribe the drug chosen for the treatment of the iuxta-femoral or iuxta-popliteal superficial vein thrombosis at.

**Figure 3 medicina-59-01068-f003:**
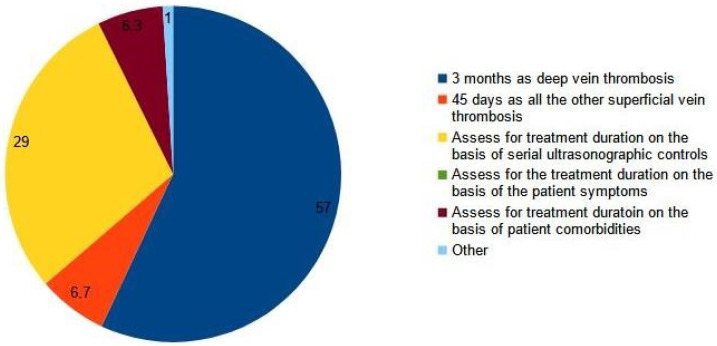
Q10: According to your experience, what should be the optimal duration of the treatment for iuxta-junctional superficial vein thrombosis?

**Table 1 medicina-59-01068-t001:** Questionnaire administered to the participants.

Q1	What is your specialization?
Q2	How many cases of superficial vein thrombosis do you see and treat annually?
Q3	How many of these superficial venous thromboses affect the last 3 cm of the great saphenous vein or the small saphenous vein or extend close to the sapheno-femoral or sapheno-popliteal junctions?
Q4	What is the first-choice pharmacological treatment for superficial vein thrombosis that is constantly adopted in the Center where you work?
Q5	What is the optimal duration of the pharmacological treatment you use in the treatment of superficial vein thrombosis?
Q6	With regard to the drug you use in the treatment of superficial vein thrombosis, how do you use the drug?
Q7	Is the therapeutic regimen you use prescribed to all patients, regardless of comorbidities (e.g., known thrombophilia, cancer, previous VTE, etc.), or is it personalized on the basis of any concomitant disease with a strong thrombogenic drive that may be present?
Q8	Which is your therapeutic strategy for the treatment of superficial vein thrombosis involving the last 3 cm of the great saphenous vein or of the small saphenous vein and extended close to the sapheno-femoral or sapheno-popliteal junctions, respectively?
Q9	You prescribe the drug chosen for the treatment of the iuxta-femoral or iuxta-popliteal superficial vein thrombosis at:
Q10	According to your experience, what should be the optimal duration of the treatment for iuxta-junctional superficial vein thrombosis?

**Table 2 medicina-59-01068-t002:** Affiliation and specialty of the participant members.

Angiology	34.54%
Vascular Surgery	23.71%
Internal Medicine	29.38%
Hematology	1.03%
Other	11.34%

## Data Availability

Data supporting reported results can be found in the SIAPAV Secretariat database.

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
