# Peer review of "Common Practice in the Treatment of Superficial Vein Thrombosis Involving the Sapheno-Femoral Junction: Results from a National Survey of the Italian Society of Angiology and Vascular Medicine (SIAPAV)"

_medicina, 2023, doi:10.3390/medicina59061068_

Round 1

Reviewer 1 Report

Dear Editors,

I am sending you a list of comments with the aim to make this paper better.

1. There are no results in abstract?. The authors should include the most important results in the abstract section.

2. In the questionnaire there are no questions that answers the most important question and that is "why". For example, why is heterogeneity in the overall management od SVT present? Because of that i recommend to author to identify some socio-demographic or other characteristics of respondents and use them as independent variables to identify do they have an affect on differences in common practice in the treatment of SVT involving SFJ.

3. In the results question authors shoul present results of all asked questions, not just two of them. You can present the answers in a table.

English grammar seems fine.

Author Response

We wish to thank the reviewer for his/her comments. Hereafter our reply:

1.There are no results in abstract?. The authors should include the most important results in the abstract section.

Thank You for your suggestion, however it is quite difficult to report in the abstract the main results of the survey, because they are heterogenous. We have highlighted this fact in the abstract (labeled in red in the new version of the manuscript) and the reader is invited to read the paper to better understand this issue

2. In the questionnaire there are no questions that answers the most important question and that is "why". For example, why is heterogeneity in the overall management od SVT present? Because of that i recommend to author to identify some socio-demographic or other characteristics of respondents and use them as independent variables to identify do they have an affect on differences in common practice in the treatment of SVT involving SFJ.

Thank You for the comment. This is the main result and the object of the survey. We wish to confirm that there is a lot of confusion in approaching these patients because ther are not clear guidelines on "what-to-do" and we will use this "confusion" in order to design a prospective study that will try to give an answer to this issue. We do not need to identify peculiar characteristics of the respondents because their heterogeneity listed in table 2 (angiologists, vascular surgeons, internists, hematologists and so on) and the heterogeneity of their responses reflects exactly what we want to obtain form this survey and confirm the necessity of a specific study

3. In the results question authors shoul present results of all asked questions, not just two of them. You can present the answers in a table.

Thank You for your comment. We decided to present only the answers specific for the issue we were investigating, that is the management of the SVT close to the sapheno-femoral and sapheno-popliteal junction. Anyway, if the editor prefers we can include the answers of all asked questions as supplementary material. Let us know about your preference

Reviewer 2 Report

The authors chose to have created a questionnaire on the practice of an important topic that represents a gap of knowledge in the literature. The questionnaire did not distinguish between SVT which is close to the saphenofemoral junction and SVT in proximity to the popliteal vein. There might be a difference in clinical approach and treatment. I think this should be mentioned as a big limitation of the questionnaire and also should be taken into consideration while planning your future study. Specific comments : Introduction Line 51-51: the incidence of DVT PE is within 3 months of SVT diagnosis and not necessarily at the exact time of SVT. Line 69-70: add the percentage of patients that were treated with each approach in the RIETTE registry paper Figures 2 and 3 overlap. (not possible to see Figure 2) Make sure to publish with colors, when printed in black and white – it is not possible to distinguish between colors. Discussion: Please add that most (89% ) physicians consider these events as DVT. however, only 57% treat it as DVT ( for 3 months). Line 166 : "provide any suggestion" Did you mean "did not provide any suggestions? " Conclusion: Your conclusion relates to SVT close to iuxta- the femoral segment and not to those close to the popliteal vein. You need to be consistent throughout the document.

Author Response

We want to thank the reviewer for his/her comments. Hereafter our reply (all the amendments of the text are labeled in red): 

The authors chose to have created a questionnaire on the practice of an important topic that represents a gap of knowledge in the literature. The questionnaire did not distinguish between SVT which is close to the saphenofemoral junction and SVT in proximity to the popliteal vein. There might be a difference in clinical approach and treatment. I think this should be mentioned as a big limitation of the questionnaire and also should be taken into consideration while planning your future study.

Thank You for your comment. The goal of our questionnaire simply was to obtain information on what is the current therapeutic approach in patients with SVT which is close to the saphenofemoral junction and SVT in proximity to the popliteal vein. This is not a limitation of the survey because we really wanted this information just to know how the colleagues behaved in this situation. We agree that we will have to take this into consideration in planning the future study and that there might be a difference in clinical approach and treatment. Infact, in the upcoming study we decided to prospectively investigate  only the SVT  close to the saphenofemoral junction, leaving the SVT close to the sapheno-popliteal junction for another study. But to make this choice we needed to know how colleagues behaved in front of these patients

Specific comments : Introduction Line 51-51: the incidence of DVT PE is within 3 months of SVT diagnosis and not necessarily at the exact time of SVT.

Thank You. We amended the statement 

Line 69-70: add the percentage of patients that were treated with each approach in the RIETTE registry paper

Thank You. We added the percentages 

Figures 2 and 3 overlap. (not possible to see Figure 2) Make sure to publish with colors, when printed in black and white – it is not possible to distinguish between colors.

Thank You. We apologize for the graphical problem. Now both figures have been separated. In the template of Medicina all figures are colored. I think that when published the colors will remain, but this is an editorial issue. 

Discussion: Please add that most (89% ) physicians consider these events as DVT. however, only 57% treat it as DVT ( for 3 months).

Thank You. We amended the text 

Line 166 : "provide any suggestion" Did you mean "did not provide any suggestions? "

Thank you. We apologize for mistake, the text has been amended  

Conclusion: Your conclusion relates to SVT close to iuxta- the femoral segment and not to those close to the popliteal vein. You need to be consistent throughout the document

Thank You. We are consistent with our conclusion because we are consistent with the title of the manuscript and in the whole text we deal only with iuxta-femoral SVT, not mentioning SVT close to the sapheno-popliteal junction. Again...yes, we asked to the affiliates about its management, but our choice was directed to the approach of the iuxta-femoral SVT because it will be the object of the upcoming prospective study 

Round 2

Reviewer 1 Report

The authors explained in a detail everything that was bothering me with the results of this paper.